# Alchemical Design of Pharmacological Chaperones with Higher Affinity for Phenylalanine Hydroxylase

**DOI:** 10.3390/ijms23094502

**Published:** 2022-04-19

**Authors:** María Conde-Giménez, Juan José Galano-Frutos, María Galiana-Cameo, Alejandro Mahía, Bruno L. Victor, Sandra Salillas, Adrián Velázquez-Campoy, Rui M. M. Brito, José Antonio Gálvez, María D. Díaz-de-Villegas, Javier Sancho

**Affiliations:** 1Departamento de Bioquímica y Biología Molecular y Celular, Facultad de Ciencias, Universidad de Zaragoza, 50009 Zaragoza, Spain; mcondeg@unizar.es (M.C.-G.); juanjogf@unizar.es (J.J.G.-F.); mariagaliana8@gmail.com (M.G.-C.); amahia@unizar.es (A.M.); sandrasalillasberges@gmail.com (S.S.); adrianvc@unizar.es (A.V.-C.); 2Biocomputation and Complex Systems Physics Institute (BIFI)-GBsC-CSIC Joint Unit, Universidad de Zaragoza, 50018 Zaragoza, Spain; 3Coimbra Chemistry Center-Institute of Molecular Sciences (CQC-IMS), Department of Chemistry, University of Coimbra, 3004-535 Coimbra, Portugal; blvictor@fc.ul.pt (B.L.V.); rbrito@ci.uc.pt (R.M.M.B.); 4Aragon Health Research Institute (IIS Aragón), 50009 Zaragoza, Spain; 5CIBER de Enfermedades Hepáticas y Digestivas CIBERehd, Instituto de Salud Carlos III, 28029 Madrid, Spain; 6Instituto de Síntesis Química y Catálisis Homogénea (ISQCH), Departamento de Química Orgánica, Facultad de Ciencias, Universidad de Zaragoza, 50009 Zaragoza, Spain; jagl@unizar.es

**Keywords:** phenylketonuria, pharmacological chaperones, lead optimization, alchemical free energy calculations, binding energetics

## Abstract

Phenylketonuria (PKU) is a rare metabolic disease caused by variations in a human gene, PAH, encoding phenylalanine hydroxylase (PAH), and the enzyme converting the essential amino acid phenylalanine into tyrosine. Many PKU-causing variations compromise the conformational stability of the encoded enzyme, decreasing or abolishing its catalytic activity, and leading to an elevated concentration of phenylalanine in the blood, which is neurotoxic. Several therapeutic approaches have been developed to treat the more severe manifestations of the disorder, but they are either not entirely effective or difficult to adhere to throughout life. In a search for novel pharmacological chaperones to treat PKU, a lead compound was discovered (compound IV) that exhibited promising in vitro and in vivo chaperoning activity on PAH. The structure of the PAH-IV complex has been reported. Here, using alchemical free energy calculations (AFEC) on the structure of the PAH-IV complex, we design a new generation of compound IV-analogues with a higher affinity for the enzyme. Seventeen novel analogues were synthesized, and thermal shift and isothermal titration calorimetry (ITC) assays were performed to experimentally evaluate their stabilizing effect and their affinity for the enzyme. Most of the new derivatives bind to PAH tighter than lead compound IV and induce a greater thermostabilization of the enzyme upon binding. Importantly, the correspondence between the calculated alchemical binding free energies and the experimentally determined ΔΔ*G*_b_ values is excellent, which supports the use of AFEC to design pharmacological chaperones to treat PKU using the X-ray structure of their complexes with the target PAH enzyme.

## 1. Introduction

Phenylketonuria (PKU) is an inborn error of metabolism caused by more than 1200 different variants of the *PAH* gene, many of them leading to a reduced enzymatic activity of the encoded phenylalanine hydroxylase (PAH) enzyme (*PAH*vdb, http://www.biopku.org, last accessed on 7 February 2022). As a consequence, high phenylalanine blood levels build up, which are toxic for the brain [1]. PKU patients are classified according to their blood phenylalanine levels in three groups exhibiting phenotypes of increasing severity, namely, mild hyperphenylalaninemia, mild PKU and classic PKU [2,3]. The neurological symptoms associated with untreated PKU are mental retardation and developmental problems. Fortunately, an early diagnosis from generalized neonatal screening followed by prompt dietary intervention effectively avoids the most severe disease outcomes [4]. The mainstay of PKU treatment is a low phenylalanine diet [5] and, for milder PKU patients, supplementation with BH_4_ [6], the cofactor of the enzyme, as this compound acts as a pharmacological chaperone (PC) partly recovering the lost enzymatic activity [7]. Despite these treatments, new therapies for different PKU phenotypes are needed [8]. Enzyme replacement therapy based on a PEG-coated phenylalanine metabolizing enzyme of bacterial origin is a novel approach, but substantial side effects have already been reported upon subcutaneous administration [9,10].

In a previous work, a compound (named IV) was identified as promising PC alternative to BH_4_ in the treatment of PKU [11]. The compound showed a stabilizing effect on tetrameric wild-type PAH and several PKU variants, and gave rise to increased PAH activity in cells transiently transfected, and in mouse liver after oral administration. Compound IV, which will be referred to from here on as **IV^PC^**, was described to stabilize the tetrameric functional form of the enzyme in vitro and appeared to act as a canonical PC, binding to the PAH tetrameric folded state, displacing the folding equilibrium towards the native form and rescuing its physiological function [11]. The crystallographic structure of human PAH in complex with **IV^PC^** [12] indicates this compound binds to the active site of the enzyme, participating in the catalytic metal coordination sphere (Figure 1), which explains its behavior as a weak competitive PAH inhibitor [11]. On the other hand, PAH refolding kinetics demonstrated an additional chaperoning role for **IV^PC^**: it accelerates the folding reaction by stabilizing partly folded species transiently accumulating along the PAH-folding pathway [12]. Interestingly, the reported X-ray structure of the PAH-**IV^PC^** complex opens a way to the rational design of improved PCs for PKU treatment.

Aiming at this, we use here the X-ray structure of the complex to design reasonable **IV^PC^**-variants compatible with tight binding and to perform alchemical free-energy calculations [13,14,15] to compute the change in the affinity of their complexes, relative to that of **IV^PC^**. We have synthesized seventeen such derivatives, spanning a calculated ΔΔ*G*_b_ of 2 kcal mol^−1^ (unsigned). Their actual ΔΔ*G*_b_ and mid-denaturation temperature change (ΔT_m_) relative to the reference PAH-**IV^PC^** complex were determined using ITC experiments and thermal unfolding, respectively. The feasibility of this approach to obtain PCs with improved binding affinity for the target enzyme was assessed.

## 2. Results and Discussion

### 2.1. Design of Novel IV^PC^ Analogues as Potential Pharmacological Chaperones with New Chemical Properties

**IV^PC^**, i.e., 5,6-dimethyl-3-(4-methyl-2-pyridinyl)-2-thioxo-2,3-dihydrothieno[2,3-d] pyrimidin-4(1*H*)-one, is the lead PC of a second generation of designed analogues (compounds **IVx**). The X-ray structure of the complex between PAH and **IV^PC^** (Figure 1) [12] was used to rationally design analogues with novel substituents, aiming at increasing the affinity of the complex.

From a thermodynamic point of view, the affinity of a compound for a protein can be increased either by improving its interactions with the protein or by decreasing its interactions with the solvent. Based on the crystallographic structure of the PAH-**IV^PC^** complex, we designed chemical modifications in the three heterocycles (thiophene, pyridine and pyrimidine) of **IV^PC^** that introduce new substituents pointing toward either small cavities neighboring bound compound **IV^PC^** or to the solvent. The proposed modifications sought to enhance ligand/protein van der Waals interactions and the entropic stabilization of the complex through increased hydrophobic effect. In total, we designed and synthesized 17 **IV^PC^** analogues (Table 1) with different physicochemical and lipophilic properties (Appendix A). The chemical modifications in **IV^PC^** analogues include addition of polar groups to establish new interactions with PAH residues and addition of apolar groups (or substitution of the pyridine ring by other apolar ring systems) to fill parts of the large groove conforming the catalytic site, which remains accessible to the solvent after compound binding. One of these fillable spaces appears just below the pyridine ring (as per the spatial orientation given in Appendix A) packing onto the α-helix depicted on the right side wall of the catalytic site. Another one appears behind the coordinated center, in the direction pointed by the sulfur atom of the 2-thioxo group (in the pyrimidine central ring), and a third one appears in the direction pointed by the most buried methyl group out of the two in the thiophene ring.

Overall, the modifications lead to analogues (Appendix A) more hydrophilic than **IV^PC^** (e.g., **IVa**, **IVb**, **IVd** or **IVg**) and others more hydrophobic (e.g., **IVh**, **IVj**, **IVm**, **IVn** or **IVo**). In general, the **IV^PC^** analogues maintain the adequate drug likeness of the lead for oral administration (Lipinski’s [16], Veber’s [17] and Egan’s [18] rules), and, according to the BOILED-Egg predictive model (Appendix A), they retain the low probability of crossing the blood–brain barrier exhibited by the lead, which is convenient as they are wished to act in the liver and kidney. A few analogues, though, (**IVc**, **IVd**, **IVh**, **IVi**, **IVj** and **IVq**) may show suboptimal intestinal absorption.

### 2.2. Synthesis of IV^PC^ Analogues IVa–IVq

Analogues **IVa**–**IVg** in which substituents on the thiophene or/and the pyridine ring of the lead compound have been varied were prepared from cyclization of the corresponding mixed thioureas using conditions for *tert*-butoxide-assisted amidation of esters [19] and subsequent side chain transformation when necessary, as detailed below. 

Reaction of 2-aminothiophene **1** or **2** with 1,1′-thiocarbonylbis(pyridin-2(1*H*)-one) [20,21] in dry dichloromethane at room temperature provided thiophene-2-isothiocyanate **3** or **4** with a 69 and 85% yield, respectively. Mixed thioureas **7** and **8** were obtained with excellent yield (98–92%) by reaction of thiophene-2-isothiocyanate **3** with 4-methyl-pyridin-2-amine **5** or 4-(((tert-butyldimethylsilyl)oxy)methyl)pyridin-2-amine **6**. The reaction of thiophene-2-isothiocyanate **4** with 4-(((tert-butyldimethylsilyl)oxy)methyl)pyridin-2-amine **6** cleanly afforded mixed thiourea **9** with a 90% yield (Figure 1).

Mixed thioureas **13**–**15** were synthesized through a one pot procedure from 4-substituted pyridin-2-amines (Figure 2). Coupling was initialized by reaction of 1,1′-thiocarbonylbis(pyridin-2(1*H*)-one) with the corresponding pyridin-2-amine. After the time required for in situ generation of the pyridin-2-isothiocyanate, 2-amino-thiophene **2** was added to the resulting reaction mixture. In this way, mixed thioureas **13**–**15** were obtained starting from 4-chloropyridin-2-amine **10**, 4-methoxypyridin-2-amine **11** and 4-cyanopyridin-2-amine **12** with a 92, 85 and 88% two-step yield, respectively. This one pot procedure did not lead to the expected results when pyridin-2-amine **5** or **6** with an inductive electron-donating group at C_4_ was used as starting material.

Cyclization of the mixed thioureas **7**–**9**, **13**–**15** using potassium *tert*-butoxide in *tert*-butanol under reflux conditions provided the corresponding 3-(pyridin-2-yl)-2-thioxo-2,3-dihydrothieno[2,3-d]pyrimidin-4(1H)-ones (Figure 3). The yield obtained depended on the substitution pattern: **IVc** (63%), **16** (95%), **17** (51%), **IVe** (61%), **IVf** (74%) and **IVg** (74%).

Subsequent transformation of thiophene and/or pyridine side chain in compounds **IVc**, **16** and **17** provided analogues **IVa**, **IVd** and **IVb**, respectively.

Analogue **IVa** was obtained in 31% yield by reduction with LiAlH_4_ of the ethoxycarbonyl group at C_6_ in analogue **IVc** (Figure 4).

Removal of *tert*-butyldimethylsilyl group in compound **17** using tetrabutylammonium fluoride in THF cleanly afforded analogue **IVb** in 79% yield (Figure 5).

Reduction of the ethoxycarbonyl group at C_6_ in compound **16** with LiAlH_4_ followed by removal of *tert*-butyldimethylsilyl group in the obtained compound **18** using tetrabutylammonium fluoride in THF provided analogue **IVd** with a 49% overall yield over two steps (Figure 6).

Analogues **IVh**–**IVk** modified in the pyrimidine ring were prepared by *S*-Alkylation of analogue **IVc** with the corresponding alkyl halide in the presence of potassium hydroxide [22] (Figure 7). In this way, S-benzylation of **IVc** provided **IVh** with 92% yield, reaction of **IVc** with iodoacetonitrile led to **IVi** in 91% yield and S-allylation reaction of **IVc** with 3-chloro-2-methyl-1-propene and allyl bromide provided **IVj** and **IVk** with 84 and 91% yield, respectively.

In addition, analogue **IVl** also modified in the pyrimidine ring was prepared in 95% yield by reduction of the carboxyethyl group at C_6_ in chaperone **IVk** with LiAlH_4_ (Figure 8).

Sodium hydride promoted reaction of 2-aminothiophene **2** with aryl isothiocyanates led to analogues **IVm**–**IVo**, in which pyridine ring was replaced by another aromatic ring in a single-step process (Figure 9). The reaction with 1-isothiocyanato-3,5-dimethylbenzene, 1-isothiocyanatonaphthalene and 5-isothiocyanato-1,2,3,4-tetrahydronaphthalene cleanly afforded compounds **IVm**, **IVn** and **IVo** in 79, 80 and 74% yield, respectively.

Reaction of analogue **IVn** with iodoacetonitrile in the presence of potassium hydroxide provided analogue **IVp** with 67% yield (Figure 10).

The synthesis of analogue **IVq** started with the preparation of thiophene **19** in 31% yield by cyclization of ethyl-2-oxobutyrate and ethyl cyanoacetate with elemental sulfur according to Gewald procedure [23,24] using diethylamine as base and ethanol as solvent to avoid side chain reactions. The thiophene **19** was converted into isothiocyanate **20** with a 67% yield by reaction with 1,1′-thiocarbonylbis(pyridin-2(1*H*)-one) as described above. Isothiocyanate **20** reacted with 4-methylpyridin-2-amine **5** to afford mixed thiourea **21** in 95% yield. Finally, analogue **IVq** was obtained in 85% by treatment of mixed thiourea **21** with potassium *tert*-butoxide in *tert*-butanol (Figure 11).

### 2.3. In Silico Calculation of the Affinity of the PAH-IV^PC^ and PAH-IV_x_ Complexes

The difference in binding free energy between the PAH-**IV^PC^** complex and any of those formed by the enzyme with the **IV^PC^** analogues was determined by alchemical free-energy calculations relying on short 5 ns H-REMD simulations, as described in Methods. Two binding scenarios were considered where either a ferrous or a ferric cation (previously parameterized as described in Methods) appears coordinated to the PAH catalytic triad (residues H285, H290 and E330). Using the calculated free energies obtained from the alchemical transformations simulated on the compounds (**IV^PC^**^→^**IVx**) when bound to PAH (∆∆*G*_bound_^IVPC^^→IVx^) and when solvated alone (∆*G*_solv_^IVPC^^→IVx^), the relative binding free energy of each PAH-**IVx** complex relative to the PAH-**IV^PC^** one (∆∆*G*_b_^IVPC^^→IVx^ = ∆*G*_b_^IVx^–∆*G*_b_^IVPC^) was calculated (Table 2). The short simulation time (5 ns) setup used here for the simulations proved to be enough to obtain reproducible results with the number of replicas run (from 3 to 6). In longer simulations, a higher number of replicas led to separation of the compound from the metal center (not shown), which had to be discarded. Clearly, the differential binding energies pertaining to transformations of complexes bearing FeII show much better correspondence with the experimentally determined ones (Table 2) and are subsequently presented. Irrespective of the modeled iron redox state, complexes involving **IV^PC^** analogues carrying modifications in the iron coordinating 2-thioxo group of the pyrimidine ring (analogues **IVh**, **IVi**, **IVj**, **IVk**, **IVl** and **IVp**) resulted in poorly reproducible trajectories, with the ligand often being displaced from the enzyme binding site and losing its coordination with the metal center. For some of these complexes we could still determine ∆∆*G*_b_**^IVPC^**^→^**^IVx^** values by increasing the number of simulation replicas and selecting those where the ligand remained bound at the original site at the end of the alchemical transformation. This compound disconnection from the metal center was not observed in the other simulated complexes.

For the complexes bearing FeII, negative ∆∆*G*_b_^IVPC^^→^^IVx^ values (meaning increased affinity) were calculated for analogues **IVa**, **IVc**, **IVe**, **IVf**, **IVm**, **IVn**, **IVo** and **IVq**, while analogues **IVb**, **IVd** and **IVg** were calculated to form just slightly less tight complexes than **IV^PC^** (Table 2). Thus, unlike the substitutions at the 2-thioxo group in the central pyrimidine ring, which are calculated to be destabilizing, the single substitutions introduced at the thiophene ring and the nonpolar substitutions done at the pyridine ring are calculated to either significantly increase the affinity of the analogue for the protein or, in a few cases, to only mildly decrease it.

### 2.4. Actual Affinity of the PAH-IVx Complexes and its Effect on PAH Thermostability

The thermostabilizing effect of **IV^PC^** analogues on PAH was determined by monitoring PAH unfolding curves using Trp emission fluorescence. In the initial screening work where **IV^PC^** was discovered, its PAH stabilizing effect was assessed by fitting the fluorescence curves to a two-state unfolding model [11]. In a recent work describing PAH thermal unfolding in more detail, two spectroscopic thermal unfolding transitions were noticed, and the unfolding curves were fitted to a three-state model [25]. As **IV^PC^** primarily stabilizes the second unfolding transition of PAH, the one that takes place at a higher temperature and gives rise to the larger emission intensity, and as **IV^PC^** analogues appear to do the same, we evaluated, for simplicity, their thermostabilizing effects by fitting the corresponding unfolding curves to the simpler two-state model (Figure 2). Thus, Δ*T*_m_ values (*T*_m_^IVx^–*T*_m_) are reported, which essentially coincide with Δ*T*_m2_ values.

Only three analogues carrying substitutions in the thioxo group (**IVh**, **IVj** and **IVk**) show a thermodestabilizing effect, suggesting they could preferentially bind to the PAH unfolded state rather than to the folded one (Figure 2a). Analogues **IVb**, **IVf** and **IVl** exert a moderate stabilizing effect (0 < Δ*T*_m_ < 5 °C), which is below that of the lead **IV^pc ^**(Table 2 and Figure 2a). All the other analogues stabilize the enzyme more than **IV^pc^** (Figure 2b). Compounds **IVi**, **IVm**, **IVn** and **IVp** show an impressive effect as they increase the *T*_m_ by more than 10 °C.

Thermostabilization of a protein because of ligand binding tends to correlate with the affinity of the complex the ligand forms with the native state of the protein [26]. However, the affinity of the complex is also influenced by other thermodynamic parameters, such as Δ*H*_b_, or Δ*C**p*_b_. Therefore, a perfect correlation between Δ*T*_m_ and ΔΔ*G*_b_ is not expected. To determine ΔΔ*G*_b_ experimentally, ITC assays were performed with the synthesized analogues and PAH (Table 2 and Figure 3). The affinity of all the complexes is in the micromolar range, and nine of them exhibit a higher affinity (lower *K*_d_) than **IV^PC^** (Figure 3). As anticipated, qualitative agreement is observed (Appendix A) between the Δ*T*_m_ associated to each ligand and the ΔΔ*G*_b_ determined in the **IVx** complexes relative to the complex formed by **IV^PC^**. Clearly, the more thermostabilizing analogues are among those with higher ΔΔ*G*_b_ values (Appendix A).

The effect of introducing substituents in the 2-thioxo group of the central pyrimidine ring on the affinity of the complexes can be assessed by comparing the affinity of the complexes formed by analogues **IVh**, **IVi**, **IVj** and **IVk** with that of **IVc**, the affinity of complex formed by **IVl** with that of **IVa**, and by comparing the affinity of the complex formed by **IVp** with that of **IVn**. The small and polar –CH2–CN substituent (**IVi** and **IVp**) leaves the affinity of the complexes close to that of their references (**IVc** and **IVn**, respectively, see Table 2). The two other substituents tested, –CH2–C(CH3)═CH2 (**IVj**) and –CH2–CH═CH2 (**IVk** and **IVl**) destabilize the complex, as shown by comparison with their references (**IVc, IVc** and **IVa**, respectively, Table 2), in agreement with the anomalous behavior observed in the MD simulations used to calculate the alchemical ΔΔ*G*_b_ values for those derivatives.

In contrast, the **IVa**/**IV^PC^**, **IVc**/**IV^PC^** and **IVd**/**IVb** pairs indicate that replacing the apolar 6-methyl group at the thiophene ring either with –CH_2_OH or COOEt groups increase the affinity of the complex. The affinity is also increased by replacing the pyridine ring with bulkier chemical groups (see **IVm**,**n**,**o/IV^PC^** pairs) or by substitution of the methyl group by –OCH_3_ (see **IVf**/**IV^PC^** pair) or –Cl (see pair **IVe**/**IV^PC^**), but not by -CN (pair **IVg**/**IV^PC^**), which hardly changes the affinity, or –CH_2_OH (see **IVb**/**IV^PC^** and **IVd**/**IVa** pairs) which is destabilizing. The described stabilizing substitutions indicate that both the surface-exposed thiophene ring and the deeply buried pyridine one can be substantially modified resulting in an increase of the affinity of the complex with PAH. Instead, modifications of the central pyrimidine ring may not be equally promising.

The thermodynamic profile of ligand binding gives clues on the interactions and effects contributing to the observed stability of a protein/ligand complex, such as the dominance of direct protein/ligand interactions (e.g., hydrogen bonds or van der Waals) or of nonspecific protein or ligand desolvation (e.g., hydrophobic effect). In general, the entropic optimization of ligands has proved easier to achieve than the enthalpic one due to the difficulty of adding polar groups in the ligand structure at the appropriate distance and orientation to establish strong interactions with the target [27]. The binding thermodynamic profiles of **IV^PC^** and all thermostabilizing analogues (Figure 3) were obtained from isothermal calorimetric titrations of PAH (Appendix A). The thermodynamically favorable binding of **IV^PC^** to PAH, reflected in its negative Δ*G*_b_ value, arises from favorable enthalpic and entropic contributions. Although the PAH-**IV^PC^** complex involves the formation of two hydrogen bonds between chaperone atoms and protein residues [12], most of the complex stability results from the favorable entropic contribution. Thus, desolvation entropy seems to drive **IV^PC^** binding. In the case of the analogues, the same binding scenario is observed: both the entropy and the enthalpy components are stabilizing, but the entropic component dominates, evidencing that complexes exhibiting a higher affinity that **IV^PC^** tend to benefit from a larger entropic component than that of the **IV^PC^** complex.

### 2.5. Usefulness of AFEC for the Rational Design of Better PAH Binders

To validate the AFEC methodology implemented here, we compared the alchemically calculated $\Delta \Delta G_{b}$ values for the **IVx** analogues (Table 2 and Figure 4) with those experimentally determined by ITC. As discussed above, S-alkylated analogues in the pyrimidine ring gave rise to poorly reproducible MD trajectories and were excluded from the comparison. The fitting in Figure 4, corresponding to the calculations done with the FeII-bearing enzyme, are in fine agreement with the experimental data. This indicates that, having the X-ray structure of a PC bound to the PAH active site, analogues can be designed, the affinity of which can be accurately calculated using AFEC prior to chemical synthesis.

This approach can save much synthetic effort by focusing on the synthesis and testing of the more promising analogues. As the catalytic reaction mechanism of nonheme iron pterin-dependent aromatic amino acid hydroxylases is not totally clear and the redox state of the iron atom during the enzyme catalytic cycle changes [28], we also parameterized the ion as FeIII coordinated to the catalytic triad. The AFEC relative binding energies obtained for this alternative parameterization are compared to the experimental energies in Appendix A. The agreement is clearly worse than that obtained with the FeII parameterization (see also Appendix A). Thus, of the two binding models implemented in this work, the model with the coordinated ferrous cation shows the best correlation with the experimental affinity data obtained by the ITC measurement (Figure 4).

The consistency of this AFEC methodology, as implemented here on the PAH-**IVx** complexes carrying FeII, was further tested by taking advantage of the energy relationships that are implicit in the thermodynamic cycles shown in Figure 5. As the cycles show, the transformation of **IV^PC^** into **IVd** can be split into that of **IV^PC^** into **IVa** plus that of **IVa** into **IVd** (left branch) or, alternatively, into that of **IV^PC^** into **IVb** plus that of **IVb** into **IVd** (right branch).

Thus, by using the previously calculated $\Delta \Delta G_{b}^{\;IV^{PC}\to IVd}$, $\Delta \Delta G_{b}^{\;IV^{PC}\to IVa}$ and $\Delta \Delta G_{b}^{\;IV^{PC}\to IVb}$ values for the corresponding alchemical transformations of **IV^PC^** into **IVa,b,d** (see Table 2), one can arithmetically anticipate that $\Delta \Delta G_{b}^{\;IVa\to IVd}$ and $\Delta \Delta G_{b}^{\;IVb\to IVd}$ should have values of +2.25 ± 0.58 kJ/mol and −1.49 ± 1.16 kJ/mol, respectively. To check for consistency of the overall methodology, we calculated those values from the corresponding alchemical transformations using the same parameterization and AFEC protocol. The calculated values of +2.36 ± 0.69 kJ/mol and −1.38 ± 0.21 kJ/mol, respectively, show excellent agreement with the values anticipated from the cycles (Figure 5). The good correlations observed between the computational and experimental relative binding free-energy values (Figure 4) and between the calculated values from the thermodynamic cycles expressions (Figure 5) make it possible to propose this AFEC approach as a valuable tool to ease the design of second generation PCs with improved affinity (better binders) for PAH. Concerning derivatives of **IV^PC^**, the approach can be used to anticipate the change in affinity upon introducing modifications in either the thiophene or the pyridine rings, but not in the thioxo group of the pyrimidine ring. The approach is also expected to be useful to design better binders to newly identified PAH ligands for which the X-ray structure in complex with PAH becomes available.

PCs may help recover the lost activity of a protein carrying a deleterious variation (e.g., a single amino acid replacement) by different mechanisms. The classic chaperoning mechanism consists in the PC binding to the native fraction of defective protein molecules, making some of the unfolded ones to fold, as governed by the folding equilibrium constant. To exert this effect efficiently, the higher the affinity of the PC for the native enzyme, the better. This is particularly true of allosteric PCs [29], the activity of which is not expected to be complicated by concomitant inhibition of enzyme activity but to be directly related to binding affinity. In this respect, our approach demonstrates the great potential of AFEC as a reliable design tool for raising the binding affinity of a PC for a protein when the structure of its complex with the target protein is known. By calculating whether an analogue will be either a better or a worse binder to the target protein, e.g., PAH, the collection of analogues that have to be synthesized and then tested in cell or animal models can be significantly narrowed, which is essential in order to reduce costs and time in the drug development process, particularly when it is carried out in an academic setting. However, it should be recalled that the chaperoning effect may be exerted through alternative mechanisms such as modifying folding kinetics [12] or protecting the enzyme against inactivation [30]. It should be also remembered that PCs binding at active sites may behave as enzyme inhibitors. This detrimental effect can be minimized through judicious dosing regimens [31]. On the other hand, effective in vitro chaperoning is a requirement that may or may not translate into effective in vivo chaperoning, depending on the specific pharmacokinetics and pharmacodynamics of each PC candidate. In a previous work, the lead **IV^PC^** was shown to display its effect on PAH in both cell and animal models [11]. Functional studies to test the chaperoning effect of this second generation of **IV^PC^** analogues are out of the scope of this work, but their chaperoning effect will be tested in the future to assess the impact of increasing the binding affinity of **IV^PC^** on the in vivo chaperoning potency and PKU-variant specificity of the new compounds. As parent compound **IV^PC^** is a weak competitive inhibitor of PAH [11], the new family of derivatives here described (some with tighter and some with weaker binding) may help fine tune its chaperoning and inhibitory effects on PAH.

## 3. Materials and Methods

### 3.1. Reagents and Chemicals

All reagents were of analytical grade and used as obtained from commercial sources. Thiophenes **1** and **2** and pyridin-2-amines **5**, **6**, **10**, **11** and **12** are commercially available and were acquired from AK Scientific, Inc. The other compounds were synthesized. A detailed description of the synthesis and characterization of all the compounds involved in this work is reported in Appendix A. All **IV^PC^** analogues were dissolved in 100% dimethylsulfoxide (DMSO) and stored frozen at −20 °C.

### 3.2. Parameterization of the Metal Center and IV^PC^ Analogues, and Molecular Dynamics (MD) Preparation Setup for the Alchemical (AFEC) Simulations

PAH is a tetrameric metalloenzyme carrying an iron atom per subunit. Prior to performing MD simulations of the complexes between PAH and the different **IV^PC^** analogues, the coordinated center including the iron ion and either compound **IV^PC^** or one out of their analogues (**IVx**) was parameterized. Parameterization was done by following an ad hoc methodology relying on a set of programs in the AmberTools18 package [32] (mainly MCPB.py [33]). The coordinated metal center was modeled both with the iron ion with charge 2+ (FeII) and 3+ (FeIII). The protein residues His285, His290 and Glu330 were included in the Fe coordination sphere given the proximity (< 3.0 Å) of their side chain coordinated atom (Nε2 in His, Oε2 in Glu) to the iron ion in the crystal structure of the complex [12] (Appendix A). The N-ter and C-ter atoms of those residues were capped by acetyl (ACE) or N-methylamide (NME) moieties, respectively, before running Gaussian09 [34] (DFT level: B3LYP/6-31G*) to minimize the systems. Then, force constants of the bonds and angles were updated (Z-Matrix method [35]), and the Merz–Kollman charges [36] extracted and subsequently fitted by the RESP method [37] (see final charges in Appendix A). The parameters and charges obtained were then combined with those for the rest of the protein, as taken from the Amber99SB force field [38].

On the other hand, **IV^PC^** and **IVx** analogues were parameterized by means of the DFT function B3LYP/6-31G* (bonds and angles force constants updated), the General Amber Force Field (GAFF) [39] was set for providing the remaining van der Waals (vdW) and coulombic parameters for these compounds, and the Merz–Kollman charges [36] fitted through the RESP method [37]. Since compound **IV^PC^** in complex PAH-**IV^PC^** appears directly coordinated through the nitrogen N1 of the pyrimidine central ring [12], it seems this nitrogen loses its hydrogen before coordinating the metal center (otherwise it may impede proper coordination). How this occurs, either through a tautomeric mechanism (passing this hydrogen to the 2-thioxo sulfur atom) or of a reduction inflicted by the iron or another reducing agent, is unclear and not be addressed here. In this protocol, we thus setup and parameterized compound **IV^PC^** and most of its analogues (the exceptions being **IVi**, **IVj**, **IVk**, **IVl** and **IVp**, which are all neutral; see structures in Table 1) with an overall charge of minus one (−1), which means that we removed the referred hydrogen atom from nitrogen N1 on these compounds.

For the MD simulations, the compounds (**IV^PC^** and its analogues) and their corresponding complexes (PAH-**IV^PC^** or PAH-**IVx** with either FeII or FeIII) were embedded in a 3 nm and an 8 nm diameter octahedral box, respectively, filled (solvated) with Tip3p [40] water molecules and neutralized with Na^+^ counterions (when required). The preparation phase was completed with the following sequence of steps: minimization, heating (NVT) and equilibration (NPT). Namely, a 10,000-step steepest-descent minimization was run and then the systems were heated to 300 K through a T-ladder (consisting in running 6 consecutive NVT steps of 50 ns each, with T constant over individual steps and increasing 50 K when passing to the next one, using the Berendsen thermostat [41]). This was followed by a 200 ns NVT step (at the final simulation T of 300 K) that was performed to change to the v-rescale thermostat [42], and by two NPT steps (1 atm), the first one a 250 ns step with the Berendsen barostat [41] and the second one a 250 ns step with the Parrinello–Rahman barostat [43,44,45]. A cutoff of 0.9 nm was set as the maximum radius to account for short-range vdW and coulombic interactions. For the short-range vdW interactions, a potential-shift-Verlet modifier was setup, while for the long-range vdW and electrostatics interactions a PME scheme was implemented. ”All-bonds” restraints were applied (LINCS algorithm [46]). In the AFEC productive phase (5 ns) Hamiltonian replica exchange molecular dynamics (H-REMD) simulations [47,48,49] were performed to enhance conformational sampling and ensure convergence when calculating alchemical free energies [50] for the targeted PAH binders. Sixteen lambdas (see Appendix A) were settled and optimized to turn on/off the forces acting on the atoms being transmuted. The GROMACS 4.6.1 package [51] was used to run all the MD simulations.

### 3.3. Alchemical Free Energy Calculation (AFEC)

Calculation of the relative alchemical binding free energies (∆∆*G*_b_) of the targeted **IVx** analogues versus **IV^PC^** in the PAH complex was implemented, as shown in Appendix A. AFEC transformations (**IV^PC^**→**IVx**) on solvated compounds enabled obtaining the ”solvating” free energy (∆*G*_solv_), whereas the corresponding AFEC transformations on compounds bound to the metal center in the complex allowed to extract the free-energy change for the ”bound” state (∆*G*_bound_). The later, ∆*G*_bound_, was calculated both for the systems setup with FeII and FeIII (Appendix A), and the relative ”binding” free energy ($\Delta \Delta G_{b}^{{\mathrm{IV}}^{\mathrm{PC}}\to \mathrm{IVx}}=\Delta G_{b}^{{\mathrm{IV}}^{\mathrm{PC}}}-\Delta G_{b}^{\mathrm{IVx}}$) was then calculated Equation (1) subtracting the ”solvating” free energy from the ”bound” free energy:(1)$$\Delta \Delta G_{b}^{{\mathrm{IV}}^{\mathrm{PC}}\to \mathrm{IVx}}=\Delta G_{bound}^{{\mathrm{IV}}^{\mathrm{PC}}\to \mathrm{IVx}}-\Delta G_{solv}^{{\mathrm{IV}}^{\mathrm{PC}}\to \mathrm{IVx}}$$

The multistate Bennet acceptance ratio (MBAR) method [52] was used to calculate the (alchemical) free energy differences (“solvating” and ”bound”).

### 3.4. Expression and Purification of Human Recombinant PAH

Wild-type human PAH was recombinantly expressed in *E. coli* BL21 (DE3) cells, purified as described [12,25] and obtained as a tetramer. Essentially, the enzyme was overexpressed as a fusion protein with maltose-binding protein (MBP), purified to homogeneity by affinity chromatography, cleaved from MBP, MBP removed in a second affinity step, and PAH in its tetrameric form finally recovered after a molecular exclusion chromatography step. Final fractions were analyzed by SDS-PAGE, and their concentration determined spectrophotometrically using the theoretical molar extinction coefficient [53].

### 3.5. Fluorescence Thermal Denaturation Measurements

Thermal denaturation of PAH was monitored by tryptophan fluorescence emission (λ_exc_ = 295 nm and λ_em_ = 345 nm), from 20 to 90 °C at a heating rate of 1 °C × min^−1^ on a Cary Eclipse Fluorescence Spectrophotometer (Varian). PAH samples (2 µM monomer concentration) in 20 mM Tris, pH 7.4, with 200 mM NaCl and 100 µM compound (either **IV^PC^** or **IVx**) were prepared for differential scanning fluorimetry analysis. Compounds were initially dissolved in 100% DMSO so that the final DMSO concentration in the samples was 2.5% in all cases. Controls containing PAH (2 µM monomer concentration) and 2.5% DMSO were included.

For simplicity, data analysis was performed by fitting each experimental fluorescence curve to a two-state unfolding model, as described [54], using Equation (2):(2)$$F=\frac{\left(F_{N}^{o}+m_{N}\times T\right)+\left(F_{U}^{o}+m_{U}\times T\right)\times e^{-\Delta G/\left(R\times T\right)}}{1+e^{-\Delta G/\left(R\times T\right)}}$$
where *F* corresponds to the fluorescence intensity signal at a given temperature while $\left(F_{N}^{o}+m_{N}\times T\right)$ and $\left(F_{U}^{o}+m_{U}\times T\right)$ represent the temperature-dependent fluorescence signal of the native and unfolded states of the protein, respectively. $F_{N}^{o}$ and $F_{U}^{o}$ represent the fluorescence signal of the native and unfolded protein at a reference *T* = 0 K, and $m_{N}$ and $m_{U}$ are the slopes of their linear temperature dependencies. Furthermore, ∆*G* is the unfolding Gibbs energy change, *R* is the universal gas constant and *T* is the temperature.

The fraction of unfolded protein ($\chi_{U}$) was determined using Equation (3):(3)$$\chi_{U}=\frac{F-\left(F_{N}^{o}+m_{N}\times T\right)}{\left(F_{U}^{o}+m_{U}\times T\right)-\left(F_{N}^{o}+m_{N}\times T\right)}$$
where the temperature of mid-denaturation, *T*_m_, is the temperature at which half of the protein molecules are in the unfolded state ($\chi_{U}=0.5)$ and the other half are native. Thermal shift (Δ*T*_m_) is calculated as the difference between the *T*_m_ values determined in presence and absence of compound.

### 3.6. Isothermal Titration Calorimetry (ITC)

ITC measurements were carried out in an Auto-iTC200 (MicroCal, Malvern-Panalytical, Malvern, United Kingdom) using carefully degassed ligand (**IV^PC^** or **IVx** analogues) and PAH solutions. A 300 µM solution of **IV^PC^** or analogues dissolved in PBS, pH 7.4, was titrated into PAH (20 µM monomer concentration) in the same buffer. Ligand solutions were prepared from stock compound solutions in 100% DMSO, and all working solutions contained the same residual DMSO concentration (2.5%). Binding titrations were performed at 25 °C by successive injections of 2 µL ligand solution into the reference cell every 150 s, with a stirring speed of 750 rpm. Thermodynamic parameters of the binding equilibrium (the binding constant, *K*_b_, the binding enthalpy change, ∆*H*_b_, and the binding stoichiometry, *n*) were calculated through nonlinear least squares regression analysis of the data, by using a one-site binding model implemented in the MicroCal LLC ITC module from the Origin 7.0 software package (OriginLab, Northampton, MA, USA). The binding Gibbs energy change, ∆*G*_b_, the dissociation constant, *K*_d_ and the entropic component, −*T ×* ∆*S*, were obtained from basic thermodynamic relationships with the previously calculated thermodynamic binding parameters [55].

## 4. Conclusions

AFEC makes it possible to anticipate the change in the affinity of the PAH-**IV^PC^** complex upon modification of the chemical structure of the bound pharmacological chaperone. Good correlations between calculated and experimental $\Delta \Delta G_{b}$ values are obtained by using a model with a FeII parameterized iron ion coordinating the residues of the catalytic triad. Based on this computational approach, a new generation of **IV^PC^** analogues was obtained exhibiting improved binding affinity for the target enzyme, which translates into higher PAH thermostabilization. AFEC, and the computational calculation of properties such as those available using absorption, distribution, metabolism, excretion or toxicity predictors, can play an important role in medicinal chemistry by guiding the early selection of the more promising analogues, thereby reducing the time and cost required for their synthesis and testing.

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
