# Peer review of "Alchemical Design of Pharmacological Chaperones with Higher Affinity for Phenylalanine Hydroxylase"

_ijms, 2022, doi:10.3390/ijms23094502_

Round 1

Reviewer 1 Report

The article by María Conde-Giménez is interesting in the filed of structure based drug design because it shows a good correlations between calculated and experimental ΔΔ?b values. These findings support the use of AFEC as a tool to facilitate the identification of more promising analogues in silico. 

Author Response

We thank the possitive coment by the reviewer.

Reviewer 2 Report

The manuscript “Alchemical Design of Pharmacological Chaperones with higher 2 Affinity for Phenylalanine Hydroxylase” by Sancho and coworkers concerns with the design, synthesis and in vitro characterization of molecules to be used as Pharmacologial Chaperones to treat Phenylketonuria. Actually, the authors moved from a lead compound previously identified (the compound IV) and modified it in one or more than one sites. They used alchemical free-energy calculations (AFEC) on the structure of the PAH-IV complex to design the new derivatives.

The authors describe the synthesis and the characterization of 17 analogous of compound IV. In silico and in vitro studies highlighted that some of these new compounds showed a higher affinity for the target enzyme compared to compound IV, some other showed a lower affinity.

Overall, the experiments and the results are clearly described and I think the paper should be published although I have a main concern regarding the main hypothesis pursued by the authors.

The Pharmacological Chaperone therapy is a strategy increasingly applied to treat genetic diseases when mutations cause a destabilization of the protein but do not abolish its biological function. This approach utilizes small molecules that bind the protein target and prevent its premature degradation. In so doing the intracellular concentration of the mutated protein increases.   

The author seems to move from the hypothesis that the stronger the binding, the better is the Pharmacological Chaperone. However, they did not show any data concerning the effect of the new synthesized molecules on the activity of the PAH.  Are they inhibitors?  If this were the case, the authors should consider that an ideal PC should stabilize the target but it should not severely interfere with the biological activity of the target.

In this context the case of Fabry disease is emblematic. A PC is available and it is currently used to treat patients carrying amenable mutations (DGJ). DGJ is an inhibitor and it is administered every other day in order to balance the stabilizing effect (that is required) and the inhibitory effect (that is deleterious).

In general there is a growing interest in the development of a second-generation PCs, the allosteric PCs (please see  https://doi.org/10.3390/molecules25143145 and references therein).

In the end I think the paper deserves to be published provided that this critical point is at least discussed.

Minor points:

Fig 2 panel A: it is too crowded; I suggest to split the curves on two separate panels.

Author Response

We thank the possitive comments by the reviewer.

We agree that simultaneous roles of enzyme stabilization and inhibition are exerted by PCs im mso,e cases. Indeed, we had already indicated this fact in the manuscript (see line 73 of the initial submission).

To further clarify this fact and to briefly mention alosteric PCs, as suggested by the reviewer, we have incorporated the following highlighted sentences at  the end of the Discussion.Besides, two new references have been included as per reviewer's suggestion.

...To exert this effect efficiently, the higher the affinity of the PC for the native enzyme, the better. This is particularly true of allosteric PCs [29], the activity of which is not expected to be complicated by concomitant inhibition of enzyme activity but to be directly related to binding affinity.

...through alternative mechanisms such as modifying folding kinetics [12] or protecting the enzyme against inactivation [30]. It should be also remembered that PCs binding at active sites may behave as enzyme inhibitors. This detrimental effect can be minimized through judicious dosing regimens [31].

...on the in vivo chaperoning potency and PKU-variant specificity of the new compounds. As parent compound IVpc is a weak competitive inhibitor of PAH [11], the new family of derivatives here described (some with tighter and some with weaker binding) may help fine tuning its chaperoning and inhibitory effects on PAH.

Reviewer 3 Report

In 2012 the authors reported the identification of a pharmacological chaperone named IV showing a stabilizing effect of PAH, the protein implicated in phenylketonuria. Based on the X-ray structure of the complex PAH-IV, authors proposed the design of improved pharmacological chaperones of PAH. They performed the synthesis of seventeen compounds, alchemical free energy calculations, ITC experiments and thermal unfolding studies.

The manuscript is clear and well written, except the part about the chemical synthesis which requires major modifications.

Major corrections:

In paragraphs 2.1 and 2.2, the authors deal respectively with the design of novel analogues and with the alchemical calculations. Between the two paragraphs, a section describing the synthesis of the compounds is missing. The authors have chosen to described the chemistry in the “material and methods” part. This text and the associated schemes 1-6 should be moved in the section “results and discussions”. Moreover, Yields should be added in the scheme and commented more accurately in the text.

In the supplementary information, HR-mass analyses of the final compounds IVa-p are missing: calculated and found values should be added as well as mass spectra.

Minor corrections:

Table 1: chemical names are not useful and should be removed

Figure 2b error bars are lacking

Line 63: author mentioned compound III but there is no representation of this compound.

In supplementary information (page 14), 19F-NMR spectra are mentioned but there is no fluorine in the reported compounds.

eThis manuscript described the design of seventeen novel analogues using alchemical free-energy calculations (AFEC) on the structure of the PAH-34 IV complex. The thermal shift and isothermal titration calorimetry (ITC) assays have been performed to experimentally evaluate their stabilizing effect and their affinity for the enzyme. Most of the new derivatives bind to PAH tighter than lead com-38 pound IV and induce a greater thermostabilization of the enzyme upon binding. Importantly, the 39 correspondence between the calculated alchemical binding free energies and the experimentally 40 determined ΔΔGb values is excellent, which supports the use of AFEC to design pharmacological 41 chaperones to treat PKU using the X-ray structure of their complexes with the target PAH enzyme

Author Response

We thank the reviewer for the possitive comments.

We have followed the suggestions in full.

We have moved the chemistry to Results and Discussion intercalating it beween initial sections 2.1 1nd 2.2.

Yields have also been incorporated and commented.

HR-mass analyses of the final compounds IVa-p is now presented: calculated and found values are now indicated.

Chemical names have been removes from Table 1.

Former Figure 2b error bars included (now figure 3).

We have rephrased line 62 to eliminate mention to compound III, which had been included in the initial version only for historical reasons.

The wrong mention to 19F-NMR spectra has been removed. 

Reviewer 4 Report

The authors present an interesting investigation on the possibility of using alchemical free energy calculations for the rational design pharmacological chaperones to treat Phenylketonuria, based on the binding affinity of the lead compound IV to the target PAH enzyme.

This class of compounds represents an important possibility for the treatment of a rare metabolic disease and therefore the topic is a timely one.

The computational results have been compared to experiments, finding a good correlation and the authors critically analyzed the cases and the reasons when such correlation was not expected.

The adopted computational procedure, though relatively standard, is more than adequate and has been applied (and described) very carefully, including the not-so-standard parametrization of the iron ions.

The manuscript is clear, very well written and, above all, very well discussed (though, here and there, there are some repetitions), the obtained results are valuable as they show that this approach can successfully be used to design new compounds with increased binding affinity and potentially an enhanced pharmacological activity. 

The English is generally correct, only minor misprints (for instance line 438 should be "How this does happen...") are present.

In my opinion the manuscript deserves to be published in Int. J. Mol. Sci.

Author Response

We thank the possitive comments by the reviewer.

We have corrected the indicated misprint, as suggested.

Round 2

Reviewer 3 Report

I thank the authors for their corrections and I recommend the publication of the manuscript in its corrected form.